# Guanidine production by plant homoarginine-6-hydroxylases

Dietmar Funck[1]*, Malte Sinn[1], Giuseppe Forlani[2], Jörg S Hartig[1]*

[1]Department of Chemistry, University of Konstanz, Konstanz, Germany; [2]Department of Life Science and Biotechnology, University of Ferrara, Ferrara, Italy

*For correspondence:
dietmar.funck@uni-konstanz.de
(DF);
joerg.hartig@uni-konstanz.de
(JSH)

Competing interest: The authors declare that no competing interests exist.

**Abstract** Metabolism and biological functions of the nitrogen-rich compound guanidine have long been neglected. The discovery of four classes of guanidine-sensing riboswitches and two pathways for guanidine degradation in bacteria hint at widespread sources of unconjugated guanidine in nature. So far, only three enzymes from a narrow range of bacteria and fungi have been shown to produce guanidine, with the ethylene-forming enzyme (EFE) as the most prominent example. Here, we show that a related class of $Fe^{2+}$- and 2-oxoglutarate-dependent dioxygenases (2-ODD-C23) highly conserved among plants and algae catalyze the hydroxylation of homoarginine at the C6-position. Spontaneous decay of 6-hydroxyhomoarginine yields guanidine and 2-aminoadipate-6-semialdehyde. The latter can be reduced to pipecolate by pyrroline-5-carboxylate reductase but more likely is oxidized to aminoadipate by aldehyde dehydrogenase ALDH7B *in vivo*. Arabidopsis has three 2-ODD-C23 isoforms, among which Din11 is unusual because it also accepted arginine as substrate, which was not the case for the other 2-ODD-C23 isoforms from Arabidopsis or other plants. In contrast to EFE, none of the three Arabidopsis enzymes produced ethylene. Guanidine contents were typically between 10 and 20 nmol*(g fresh weight)$^{-1}$ in Arabidopsis but increased to 100 or 300 nmol*(g fresh weight)$^{-1}$ after homoarginine feeding or treatment with Din11-inducing methyljasmonate, respectively. In 2-ODD-C23 triple mutants, the guanidine content was strongly reduced, whereas it increased in overexpression plants. We discuss the implications of the finding of widespread guanidine-producing enzymes in photosynthetic eukaryotes as a so far underestimated branch of the bio-geochemical nitrogen cycle and propose possible functions of natural guanidine production.

## eLife assessment

This **fundamental** study advances our understanding of nitrogen metabolism by identifying a new type of guanidine-forming enzyme in eukaryotes. The key claims of the article are **convincingly** supported by the data, with meticulous biochemical, cellular, and *in vivo* studies on guanidine production. The work will stimulate interest in the cellular roles of homoarginine, and, more generally, in the biochemistry and metabolism of guanidine derivatives.

## Introduction

Nitrogen availability is a limiting factor for productivity in many ecosystems and plants are the major contributors to assimilation of nitrogen into organic matter from inorganic precursors. The largest proportion of nitrogen is used in primary metabolism for the synthesis of proteins, nucleic acids, and their precursors. Additionally, many secondary metabolites that serve for internal signaling, adaptation or interactions with other organisms are nitrogen-containing molecules. Guanidine, which exclusively occurs as the protonated guanidinium cation under physiological conditions, has been discovered as a degradation product of guanine and arginine under chemically harsh conditions (*Bénech and*

*Kutscher, 1901*; *Strecker, 1861*). Thirty-two years after its discovery in 1861, guanidine was detected in plants (*Schulze, 1893*). However, guanidine has no known function in plants, and it is unclear to which extent and how plants can synthesize and metabolize this nitrogen-rich compound (*Marsden et al., 2015*). In contrast, two enzymatic pathways for guanidine catabolism were recently discovered in bacteria and many bacteria express guanidine importers or exporters in response to externally supplied or internally produced guanidine (*Funck et al., 2022*; *Nelson et al., 2017*; *Schneider et al., 2020*; *Sinn et al., 2021*; *Wang et al., 2021*). Bacterial genes involved in guanidine metabolism are often regulated by members of four independent guanidine riboswitch classes, which may indicate an ancient evolutionary origin of guanidine metabolism (*Lenkeit et al., 2020*; *Nelson et al., 2017*; *Salvail et al., 2020*; *Sherlock and Breaker, 2017*; *Sherlock et al., 2017*).

Only two enzymes are known to release guanidine from arginine in side reactions. The Ethylene Forming Enzyme (EFE) from *Pseudomonas savastanoi* converts 2-oxoglutarate (2-OG) into ethylene and this activity depends on structural changes induced by binding of arginine to the active site (*Fukuda et al., 1992*; *Martinez et al., 2017*). In a second reaction with roughly 50% turnover rate, EFE also catalyzes the hydroxylation of arginine at the C5 atom, yielding unstable 5-hydroxyarginine, which spontaneously decays to glutamate-5-semialdehyde (GSA) and guanidine. The arginine-4,5-desaturase NapI predominantly catalyzes desaturation of the C4-C5 bond but also produces a small proportion unstable 5-hydroxyarginine (*Dunham et al., 2018*). Both enzymes belong to the class of $Fe^{2+}$ and 2-OG-dependent dioxygenases (2-ODD), which use molecular oxygen to catalyze a broad range of reactions typically involving an $Fe^{IV}$-oxo-intermediate formed after the initial conversion of 2-OG to succinate (*Farrow and Facchini, 2014*; *Hausinger, 2015*; *Herr and Hausinger, 2018*). Other enzymes from this family, like VioC or OrfP, catalyze the hydroxylation of arginine at C3 or C3 and C4, respectively, without initiating guanidine release (*Chang et al., 2014*; *Yin and Zabriskie, 2004*). A third guanidine-producing enzyme, the guanylurea hydrolase GuuH from *Pseudomonas mendocina* strain GU, is from a different protein family and may have evolved recently to utilize guanylurea generated by metformin degradation (*Tassoulas et al., 2021*).

In plants, the family of 2-ODDs has diversified immensely, and it has been divided into three subfamilies according to sequence similarity and the types of reactions that the respective enzymes catalyze (*Kawai et al., 2014*; *Prescott and John, 1996*). Subfamily A comprises RNA demethylases and related enzymes, subfamily B comprises prolyl-hydroxylases that function in posttranslational protein modification, and the largest subfamily C comprises enzymes that act in the synthesis or modification of plant hormones, pigments and other secondary metabolites. Besides the 2-ODDs with known enzymatic function, subfamily C also contains many proteins of unknown function. Among these, Arabidopsis Din11 (<u>D</u>ark-<u>in</u>duced11; At3g49620) has been identified because its expression was induced by prolonged dark treatment, which leads to C-starvation and the induction of autophagy (*Fujiki et al., 2001*; *Magen et al., 2022*). Din11 orthologs are encoded in the genomes of most analyzed plants and green algae, and Arabidopsis as well as many other *Brassicaceae* express three paralogs of this protein: Din11, At3g49630 and At3g50210, among which At3g50210 seems to be the most ancestral form (*Figure 1*). In rice, the single Din11 ortholog has been named 2-ODD33 and was reported to have weak melatonin-2-hydroxylase activity (*Byeon and Back, 2015*). Because the numbering of the 2-ODDs in rice does not reflect their phylogenetic relationship and the phylogenetic study by *Kawai et al., 2014* used DOX as rather uncommon acronym for the 2-ODDs, we propose a hybrid nomenclature with 2-ODD followed by a denominator for the phylogenetic clade as defined in *Kawai et al., 2014*. According to this nomenclature, we propose 2-ODD-C23.1 as systematic name for At3g50210, 2-ODD-C23.2 for Din11 and 2-ODD-C23.3 for At3g49630.

An important N-containing class of secondary plant metabolites are non-proteinogenic amino acids, which may have metabolic, storage, defense, or signaling functions (*Bell, 2003*). Homoarginine, the C6-analog of arginine, is accumulated to high concentrations in seeds of several *Lathyrus* species, where it is assumed to be a nitrogen storage form that is not easily accessed by bacteria or insects (*Bell, 1962*; *Bell, 1980*; *Lambein et al., 1992*; *Rao et al., 1963*). Trace amounts of homoarginine were also detected in other plants, and in humans it is thought to interfere with nitric oxide production and signaling (*Koch et al., 2023*; *Lambein et al., 2019*; *Sulser and Sager, 1976*). In man and animals, homoarginine is produced from lysine and arginine in a side reaction of glycine amidinotransferase (*Choe et al., 2013*). A protein with similar activity has been isolated from *Lathyrus sativus*, but the molecular identity of this protein remains unknown and we did not detect a homolog of human

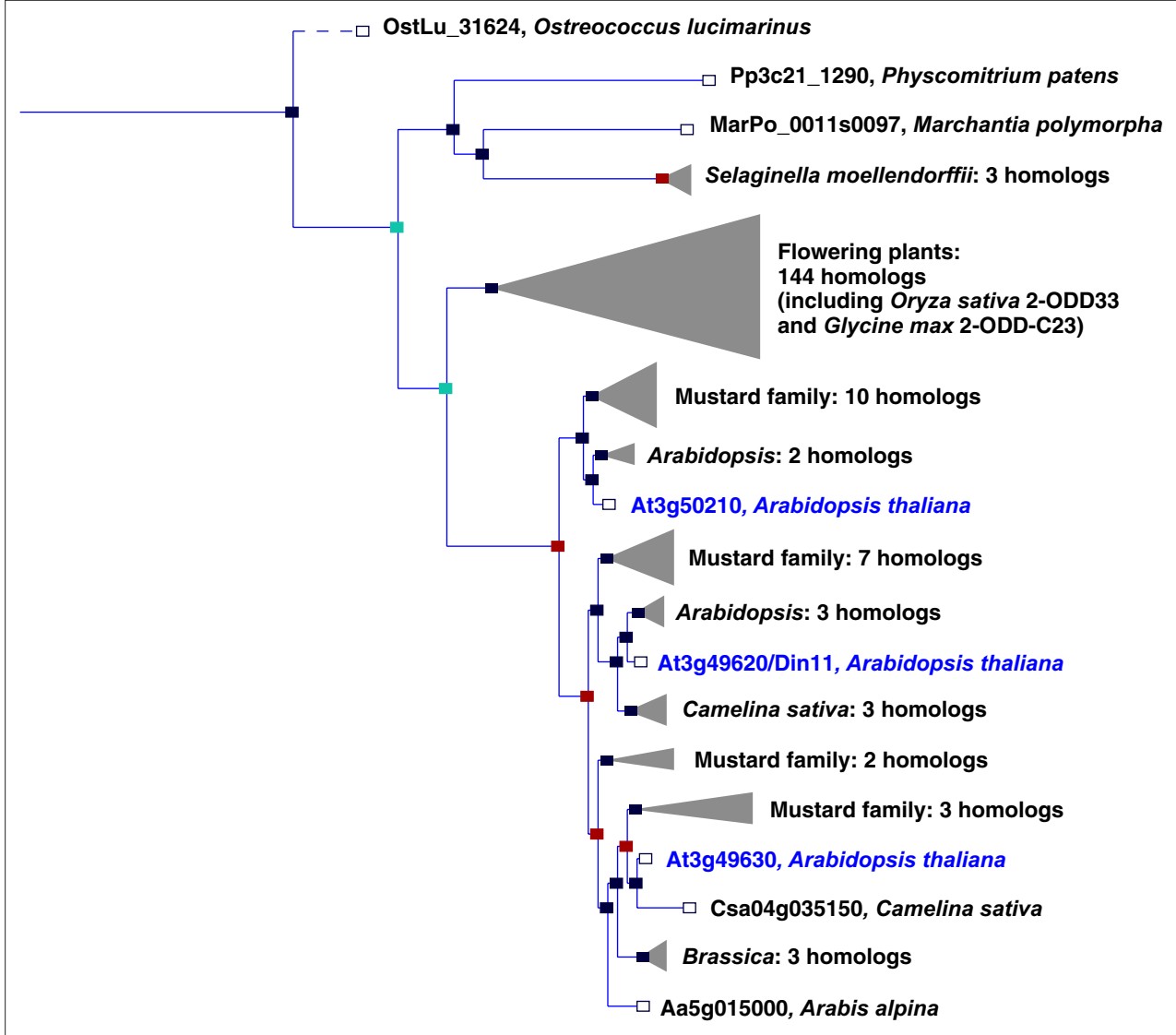

**Figure 1.** Phylogeny of the 2-ODD clade C23. Excerpt of the gene tree for *Din11* from EnsemblPlants (https://plants.ensembl.org, accessed Feb. 2023). The length of the horizontal blue lines is proportional to the number of amino acid exchanges except the dashed line, which is shortened by a factor of 10. Blue squares indicate speciation nodes, red squares indicate duplication nodes, green squares indicate ambiguous nodes, and open squares indicate species nodes. Grey triangles indicate collapsed branches. The three 2-ODD-C23 paralogues from *Arabidopsis thaliana* are highlighted by blue lettering.

The online version of this article includes the following figure supplement(s) for figure 1:

**Figure supplement 1.** Sequence comparison of structurally or functionally related 2-ODD proteins.

**Figure supplement 2.** 3'-UTR of *Din11*.

**Figure supplement 3.** Expression levels of 2-ODD-C23s under stress.

glycine amidinotransferase in any annotated plant genome (*Srivenugopal and Adiga, 1980*). No other homoarginine-metabolizing enzymes have so far been reported from plants.

Here, we describe the identification of plant 2-ODD-C23 isoforms as homoarginine-6-hydroxylases, which initiate the decay of 6-hydroxyhomoarginine to 2-aminoadipate-6-semialdehyde (AASA) and guanidine. Within the 2-ODD-C23 clade, we found Din11/2-ODD23.2 to be special because it accepted also arginine as substrate to produce 5-hydroxy-arginine. This finding identifies 2-ODD-C23s as the first eukaryotic enzymes that release free guanidine from conjugated precursors, an activity previously only known as side reaction of rare bacterial enzymes. AASA accumulation is toxic because it inactivates the enzyme cofactor pyridoxal-5-phosphate (PLP) by adduct formation. Therefore, AASA

is detoxified by oxidation to aminoadipate by ALDH7B4 (AASA-DH, At1g54100; *Končitíková et al., 2015*; *van Karnebeek et al., 2016*). In solution, AASA is in equilibrium with the cyclic δ¹-piperideine-6-carboxylate (P6C), which can be reduced to pipecolate by pyrroline-5-carboxylate reductase (P5CR; *Fujii et al., 2002*; *Struys et al., 2014*). Our comparison of the activities of P5CR and ALDH7B4 indicates that under physiological conditions the oxidation of AASA is the prevalent metabolic route in plants.

## Results

### Expression of plant EFE-like proteins causes guanidine production in *E. coli*

To determine if plant 2-ODD-C23 enzymes, which share the highest degree of similarity with EFE among the plant ODDs (26% sequence identity, *Figure 1—figure supplement 1*), also catalyze the production of guanidine, selected proteins were overexpressed in *E. coli*. We focused on the three 2-ODD-C23 isoforms from Arabidopsis (Din11/2-ODD-C23.2, At3g49630/2-ODD-C23.3 and At3g50210/2-ODD-C23.1) and the orthologs from rice and soybean, which are encoded by single-copy genes. We also included Ava5009 from *Anabaena variabilis* as representative of the most similar group of cyanobacterial 2-ODDs, although it is probably not orthologous to 2-ODD-C23 from plants and green algae. When we compared the annotation of Din11 with RNAseq data, we noticed that only very few RNAseq reads cover the 5'-end of the mRNA, whereas the coverage strongly increases 89 bp downstream of the predicted transcription start site, at a position 13 bp after the annotated start codon (*Figure 1—figure supplement 2*). A second ATG codon in the same reading frame is located 62 bp downstream of the major transcription start site. Comparison of the predicted protein sequences showed that the first 24 amino acids of the annotated Din11 (At3g49620.1, here termed

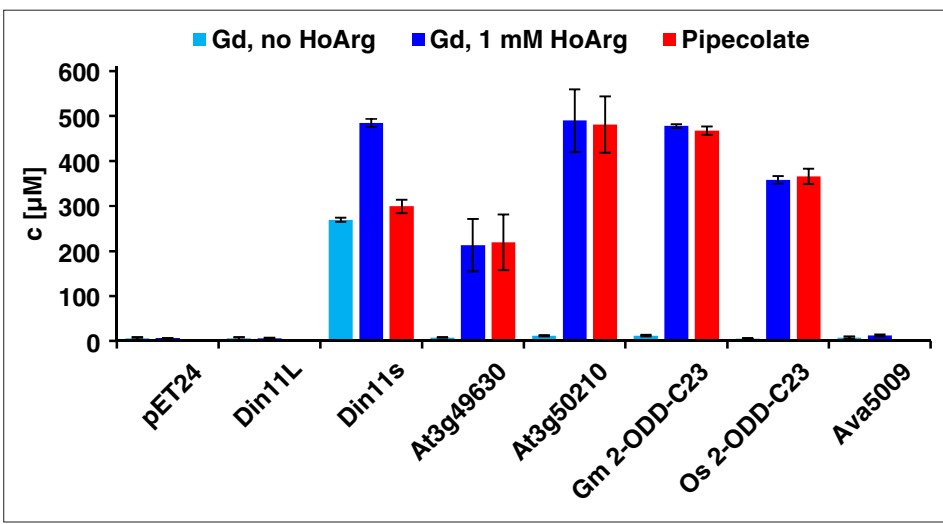

**Figure 2.** Expression of 2-ODD-C23 isoforms induces guanidine (Gd) accumulation in *E. coli*. The cDNAs of various 2-ODD-C23 isoforms from plants and Ava5009 from *Anabaena variabilis* were cloned in pET24 for overproduction of the respective proteins in *E. coli*. Expression was induced over night at 18 °C in LB medium supplemented with 0.2% (w/v) glucose and 10 µM $Fe^{2+}$. One aliquot of each culture was additionally supplemented with 1 mM homoarginine (HoArg). All cultures reached an OD around 10. For guanidine and pipecolate quantification by LC-MS, aliquots of the cultures were diluted in a 20-fold excess of methanol with 10 µM $^{13}C^{15}N$-labeled guanidine as analytical standard. Pipecolate was exclusively detected in the cultures supplemented with homoarginine. Columns are the average ± SD of samples from three independent cultures.

The online version of this article includes the following source data and figure supplement(s) for figure 2:

**Source data 1.**

**Figure supplement 1.** Distribution of metabolites in *E. coli* cultures.

**Figure supplement 1—source data 1.**

**Figure supplement 2.** 2-ODD C23 isoforms do not produce ethylene.

Din11L) do not align with 2-ODD-C23 homologs from other plant species, whereas the variant starting with the second ATG codon (termed Din11s) aligns well over its entire length. Nevertheless, we were able to amplify the coding sequence of Din11L from Arabidopsis cDNA.

Of the three Arabidopsis 2-ODD-C23 isoforms, only the short variant of Din11 induced guanidine production (detected and quantified by LC-MS, see materials and methods section) when overexpressed in *E. coli* (*Figure 2*). Guanidine production was stimulated by addition of arginine to the culture medium, whereas addition of 2-OG had no effect (*Figure 2—figure supplement 1a*). We speculated that the other 2-ODD-C23 isoforms might use a compound similar to arginine as substrate and supplemented the cultures with various other guanidine compounds known to occur in nature (methylguanidine, guanidinoacetate, guanidinopropionate, guanidinobutyrate, agmatine, homoarginine, taurocyamine). Only when the culture medium was supplemented with homoarginine, all 2-ODD-C23 isoforms from Arabidopsis except Din11L induced guanidine production in the *E. coli* cultures (*Figure 2*). The 2-ODD-C23 orthologs from soybean and rice (2-ODD33) mediated homoarginine-dependent guanidine accumulation to a similar degree than Din11s and At3g50210. The guanidine concentration was very similar in the *E. coli* cells and in the supernatant, indicating that guanidine was exported from the cells until an equilibrium with the medium was reached (*Figure 2—figure supplement 1b*). In contrast to the 2-ODD-C23 isoforms, the 2-ODD Ava5009 from the cyanobacterium *Anabaena variabilis* did not mediate guanidine production from arginine or homoarginine.

The detection of free guanidine indicated that 2-ODD-C23 might be homoarginine-6-hydroxylases, as 6-hydroxyhomoarginine is expected to decay spontaneously to yield guanidine and AASA. The latter is in spontaneous equilibrium with P6C, which can be converted to pipecolate by P5CR (proC in *E. coli*, *Fujii et al., 2002*). Indeed, we detected pipecolate in the *E. coli* cultures in quantities proportional to the amount of guanidine (*Figure 2*). Only in the cells expressing Din11s, the proportion of pipecolate was lower, suggesting that Din11s used both arginine and homoarginine as substrates. In contrast to guanidine, pipecolate was predominantly retained inside the cells (*Figure 2—figure supplement 1c*). To analyze if the plant enzymes can convert 2-oxoglutarate to ethylene, we compared bacterial cultures expressing either *Ps*EFE or one of the Arabidopsis enzymes (*Figure 2—figure supplement 2*). By GC-FID analysis of the headspace of bacterial cultures expressing EFE, Din11s, At3g49630, or At3g50210 and incubated with 1 mM arginine or homoarginine for 18 hr in air-tight tubes, we observed the formation of up to 0.18% (v/v) ethylene by cultures expressing EFE. In contrast, none of the plant enzymes produced even traces of ethylene [detection limit 0.0002% (v/v)]. As expected, the cultures expressing EFE or Din11s contained guanidine, irrespective of the supplement. The cultures expressing At3g49630 or At3g50210 only contained guanidine, when they were fed with homoarginine.

## Plant 2-ODD-C23 enzymes are homoarginine-6-hydroxylases

To determine if the Arabidopsis 2-ODD-C23 isoforms directly catalyze the release of guanidine from homoarginine, we purified the recombinant proteins by Ni-affinity chromatography. Measurement of the rate of $O_2$ consumption at variable substrate concentrations allowed the characterization of the three Arabidopsis enzymes (*Figure 3* and *Supplementary file 1*). With a fixed and saturating concentration of 0.5 mM 2-OG, At3g50210 had the lowest $K_M$ for homoarginine (0.78±0.15 mM) and the highest specific activity ($A_{max}$ = 21 ± 1.7 nmol s$^{-1}$ mg$^{-1}$). Din11s had a $K_M$ of 1.9±0.3 mM for homoarginine and a lower specific activity ($A_{max}$ = 13.4 ± 3.2 nmol s$^{-1}$ mg$^{-1}$). With a $K_M$ of 4.6±0.2 mM and an $A_{max}$ of 5.6±1.2 nmol s$^{-1}$ mg$^{-1}$, At3g49630 had the lowest activity of the three Arabidopsis 2-ODD-C23 paralogs. LC-MS/MS analysis of the reaction products confirmed the formation of P6C, succinate and guanidine. When *O*-(2,3,4,5,6-Pentafluorbenzyl)-hydroxylamine was added during the reaction, the derivatization product of AASA was also detected. No products with the masses of dehydro-homoarginine or hydroxy-homoarginine were accumulating in the reaction mixture. We concluded that all three Arabidopsis 2-ODD-C23 isoforms specifically catalyze the C6-hydroxylation of homoarginine, which is followed by the spontaneous decay to guanidine and AASA, which subsequently cyclizes to P6C. As expected from the analysis of the *E. coli* cultures, only Din11s showed $O_2$ consumption also with arginine, although the $K_M$ (6.0±5.1 mM) was higher compared to homoarginine and the specific activity was lower ($A_{max}$ = 5.2 ± 0.9 nmol s$^{-1}$ mg$^{-1}$). Neither lysine nor agmatine or any other guanidine-containing compound induced $O_2$ consumption with one of the purified enzymes. The reaction of Din11s with arginine yielded guanidine and GSA with its cyclic counterpart P5C,

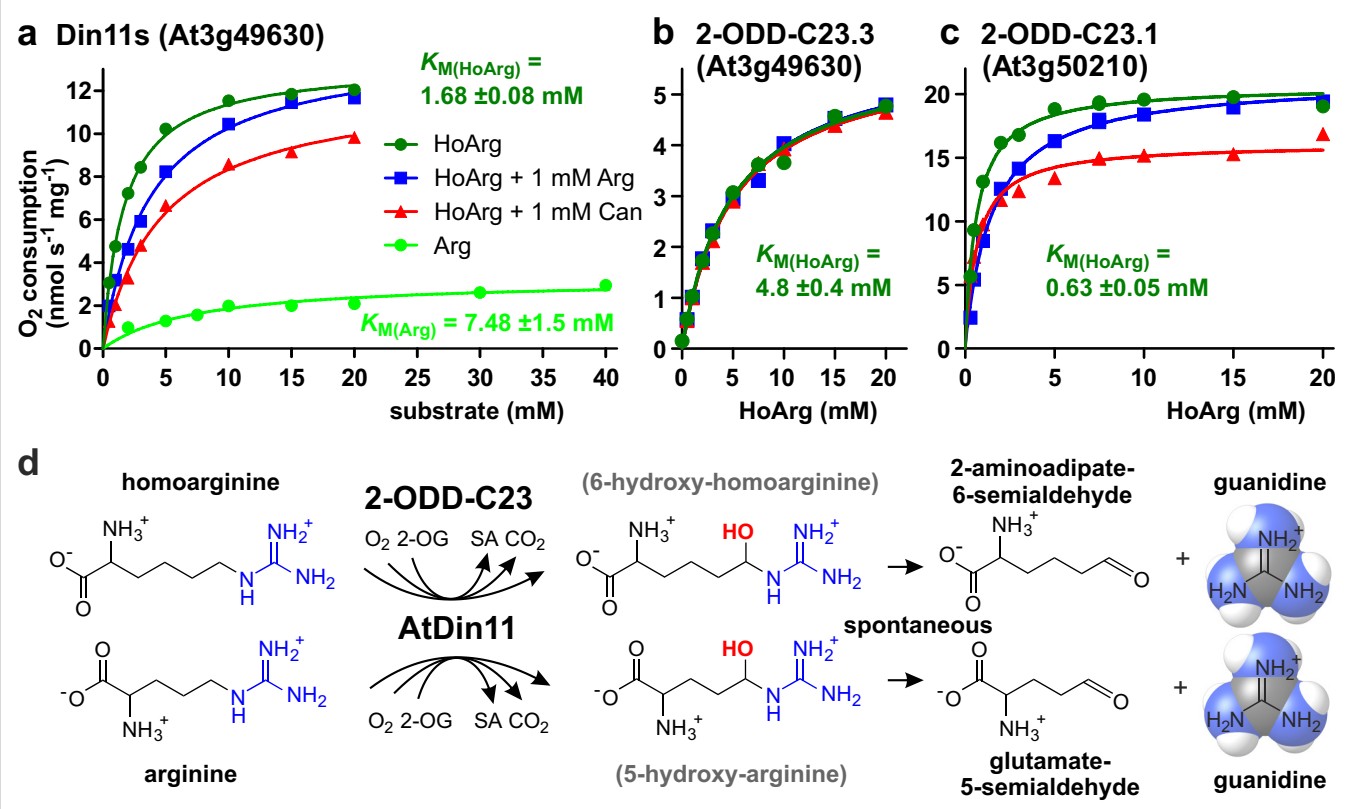

**Figure 3.** Kinetic analysis of purified recombinant 2-ODD-C23 isoforms. (**a**)-(**c**) Specific activities of oxygen consumption at various substrate and inhibitor concentrations were analyzed with a Clark-type electrode. The concentration of the co-substrate 2-oxoglutarate was fixed at 0.5 mM. Only Din11s showed oxygen consumption with both homoarginine (HoArg) or arginine (Arg). The reaction of Din11s and At3g50210 with homoarginine was inhibited in the presence of 1 mM arginine (blue lines) or canavanine (red lines). Specific activities were calculated from the slope of the initial linear reaction rate. $K_M$ and $A_{max}$ were determined by non-linear regression (see also **Supplementary file 1**). Data points represent single measurements, consistent results were obtained with independent enzyme preparations. (**d**) Reactions catalyzed by 2-ODD-C23 isoforms with subsequent spontaneous decay of the reaction products. The hydroxylated intermediates are too unstable to be detected.

The online version of this article includes the following source data and figure supplement(s) for figure 3:

**Source data 1.**

**Figure supplement 1.** Biochemical characterization of 2-ODD-C23 isoforms.

indicating that arginine is hydroxylated at the C5 position as in EFE from *Pseudomonas savastanoi* (**Fukuda et al., 1992**).

When we tested the effect of 1 mM arginine on the reaction of the 2-ODD-C23 isoforms with homoarginine, we detected a competitive inhibition of Din11s and At3g50210, whereas At3g49630 was insensitive to arginine (**Figure 3**, **Figure 3—figure supplement 1**, **Supplementary file 1**). We further tested the effect of the arginine antimetabolite canavanine on the reaction of the 2-ODD-C23 isoforms. Canavanine is the 5-oxa-analog of arginine and therefore cannot be hydroxylated at this position. Similar to arginine, which is both substrate and inhibitor for Din11s, canavanine showed competitive inhibition of the reaction of Din11s with homoarginine. In contrast, the inhibition of At3g50210 was of the mixed type and At3g49630 was not inhibited by canavanine.

When the concentration of the co-substrate 2-oxoglutarate was varied at near-saturating concentration of homoarginine (10 mM), all three 2-ODD-C23 isoforms showed $K_M$ values between 20 and 100 µM and the reaction was inhibited by high 2-OG concentrations (**Figure 3—figure supplement 1c** and **Supplementary file 1**). For At3g50210, we determined the pH-dependence of the enzymatic reaction and observed maximal activity around pH 7.2 (**Figure 3—figure supplement 1d**), whereas the enzyme was virtually inactive below pH 5 and above pH 9.

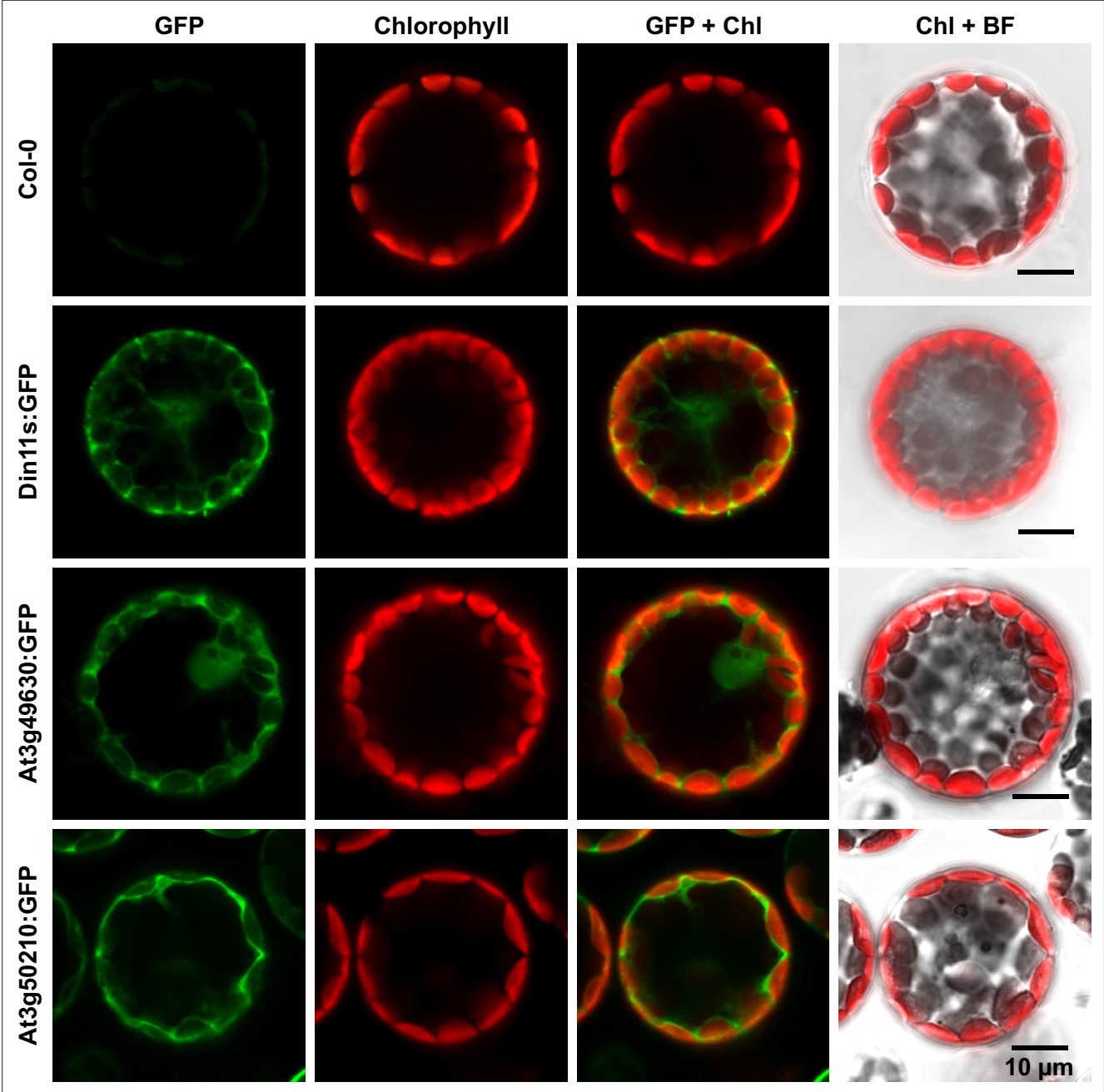

**Figure 4.** Subcellular localization of 2-ODD-C23 isoforms. Protoplasts were isolated from plants expressing 2-ODD-C23:GFP fusions under control of the constitutive UBQ10-promoter. Spectrally resolved confocal fluorescence images were used to separate GFP fluorescence (depicted in green) from chlorophyll autofluorescence (Chl, depicted in red). BF, brightfield image.

The online version of this article includes the following source data and figure supplement(s) for figure 4:

**Figure supplement 1.** Detection of GFP fusion proteins by Western blot.

**Figure supplement 1—source data 1.**

**Figure supplement 1—source data 2.**

## Arabidopsis 2-ODD-C23 isoforms are cytosolic enzymes and mediate guanidine production *in vivo*

The protein sequences of the 2-ODD-C23 isoforms from Arabidopsis do not contain any recognizable targeting signals. Consistently, the analysis of plants expressing 2-ODD-C23:GFP fusion proteins

under control of the constitutive UBQ10-promoter demonstrated the cytosolic localization of all three isoforms (*Figure 4*). Among 41 plants transformed with a construct for expression of Din11L:GFP, no plants with detectable GFP fluorescence were identified. Among 80 lines carrying a construct for expression of Din11s:GFP, several lines showed cytosolic GFP signals but only in one line, the expression of Din11s:GFP was stable over several generations. For both At3g49630:GFP and At3g50210:GFP, several independent lines were identified that stably expressed the GFP fusion protein. GFP signals were typically strongest in At3g50210:GFP-expressing plants. In soluble protein extracts, proteins of the expected size were detected with an anti-GFP antibody (*Figure 4—figure supplement 1*). Additionally, also proteins in the size of free GFP were detected in all lines, which may be derived from partial proteolysis and may explain, why GFP fluorescence was also detected in nuclei.

In order to determine whether the 2-ODD-C23 isoforms catalyze C6-hydroxylation of homoarginine under physiological conditions, we identified and generated Arabidopsis mutants. Several T-DNA insertion lines were obtained from the Nottingham Arabidopsis Stock Centre and were backcrossed at least 3 times to Col-0 wildtype (WT) plants to obtain single insertion lines (*Figure 5—figure supplement 1a* and *Supplementary file 2*). Because *Din11* and *At3g49630* are located directly next to each other on chromosome 3, we used mutagenesis with CRISPR/Cas9 with a gRNA targeting the third exon of both *Din11* and *At3g49630* to generate double mutants. We obtained one line, in which the 5'-end of *Din11* is fused to the 3'-part of *At3g49630*. Insertion of 1 bp at the fusion site causes a frame shift and inactivates both genes. A second line contains a 27 bp deletion including the border between intron 2 and exon 3 of *Din11* and a 1 bp deletion in exon 3 of *At3g49630*. The CRISPR/Cas9-generated mutants also were backcrossed to WT plants to eliminate the CRISPR/Cas9 expression cassette and potential off-target mutations. To generate triple mutants, the double mutants

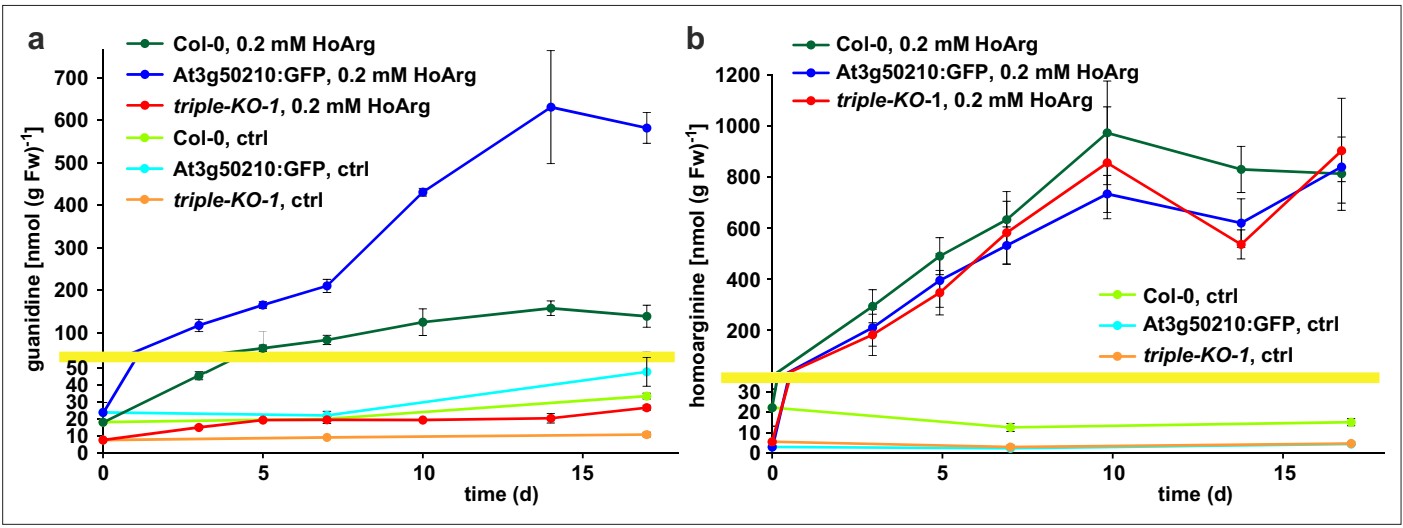

**Figure 5.** Guanidine and homoarginine contents in leaves of Arabidopsis plants. Plants were grown for two weeks in sterile culture and transferred to fresh plates with 0.2 mM or without (ctrl) homoarginine. Guanidine (**a**) and homoarginine (**b**) were extracted from rosettes in 80% (v/v) methanol containing $^{13}C^{15}N$-labeled guanidine as analytical standard and quantified by LC-MS. Col-0, wildtype; At3g50210:GFP, At3g50210:GFP overexpressing plants, triple-KO, 2-ODD-C23 triple mutant. The yellow line marks a change in the scale of the Y-axis. Error bars indicate SD from four plant samples. The entire experiment was repeated with consistent results. Continued experiments during the revision of this paper revealed that our LC-MS protocol did not separate homoarginine from methylarginine. Therefore, the identity of the compound with a m/z ratio of 189 detected in plants that were grown in the absence of homoarginine is at present ambiguous.

The online version of this article includes the following source data and figure supplement(s) for figure 5:

**Source data 1.**

**Figure supplement 1.** 2-ODD-C23 mutants in Arabidopsis.

**Figure supplement 1—source data 1.**

**Figure supplement 2.** Guanidine content in various plants.

**Figure supplement 2—source data 1.**

**Figure supplement 3.** Growth inhibition by homoarginine.

**Figure supplement 3—source data 1.**

were crossed with the mutant line *at3g50210-1*, which carries a T-DNA insertion in the third exon of At3g50210. All single, double, and triple mutants were phenotypically not different from WT plants when growing under normal conditions in sterile culture or in the greenhouse.

Basal guanidine contents were around 10–15 nmol [g fresh weight (Fw)]$^{-1}$ equivalent to 100–200 nmol (g dry weight)$^{-1}$ in the rosettes of axenically grown Arabidopsis WT plants and were increased to 20–30 nmol (g Fw)$^{-1}$ in 2-ODD-C23:GFP overexpressing plants (**Figure 5**). The guanidine content of soil grown plants was usually slightly higher, and in legume seeds and seedlings we detected up to 2.4 µmol (g dry weight)$^{-1}$ (equivalent to 145 ppm) guanidine (**Figure 5—figure supplement 2**). The guanidine content in rosettes of 2-ODD-C23-triple mutants was consistently lower than in WT plants. When WT plants and At3g50210:GFP-expressing plants were transferred to culture plates with 0.2 mM homoarginine in the medium, they accumulated both homoarginine and guanidine in a time-dependent manner, reaching 100 nmol (g Fw)$^{-1}$ guanidine in WT rosettes and 600 nmol (g Fw)$^{-1}$ in the rosettes of the overexpressor plants. Like WT and overexpressor plants, the 2-ODD-C23 triple mutants accumulated homoarginine to around 800 nmol (g Fw)$^{-1}$, whereas the guanidine content in the rosettes stayed nearly unchanged. In *din11* and *at3g49630* single mutants, the guanidine content was not reduced compared to WT plants after external supply of homoarginine. In *at3g50210* single mutants, however, the guanidine content was as low as in the triple mutants (**Figure 5—figure supplement 1b**).

Addition of homoarginine or arginine to the growth medium impaired root growth of Arabidopsis seedlings (**Figure 5—figure supplement 3a and b**). On media containing either 1 mM arginine or 0.1 mM homoarginine, the roots of 1-week-old plants had only half the length of the roots of control plants grown without amino acid supplement. Above 0.2 mM homoarginine, also the growth of the rosettes was impaired (**Figure 5—figure supplement 3c**). However, these inhibitory effects were not changed in the 2-ODD-C23 triple mutants and 2-ODD-C23:GFP overexpressing plants, indicating that hydroxylation of homoarginine likely does not serve to detoxify high concentrations of homoarginine. Addition of 1 mM guanidine to the growth medium had no effect on seedling growth, whereas 5 mM guanidine was lethal (**Figure 5—figure supplement 3d**).

Originally, Din11 was named for its induction by prolonged darkness, a condition now understood to induce both C-starvation and autophagy. However, we did not detect phenotypical differences between *din11* or triple mutants and WT plants after prolonged dark treatment (**Figure 6a**). It had

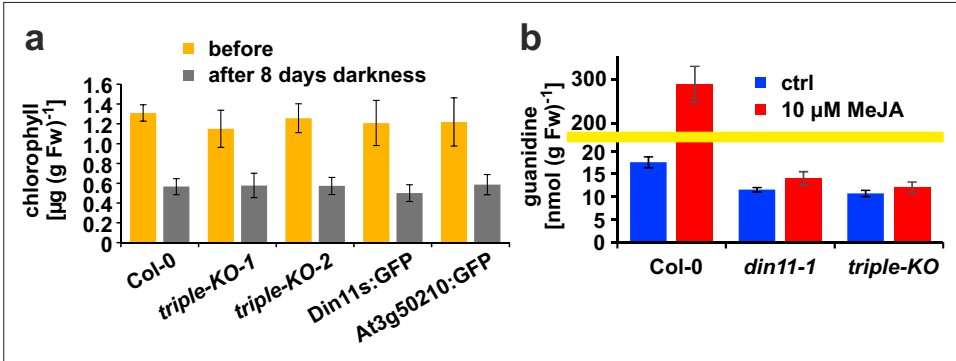

**Figure 6.** Effect of darkness and methyljasmonate (MeJA) on 2-ODD-C23 mutants. (**a**) Eight-day-old seedlings grown under short-day conditions were placed for 8 days in permanent darkness. Chlorophyll was extracted before and after the dark treatment and quantified spectrometrically. Col 0, wildtype; triple-KO-1 and 2, independently generated 2-ODD-C23 triple mutants; Din11s:GFP, At3g50210:GFP, plants overexpressing 2-ODD-C23:GFP fusion proteins. Error bars indicate SD of N=8 independent plant samples. (**b**) Three-week-old plants were treated for one week with approximately 50 µM MeJA by applying a stock solution in 50% (v/v) ethanol to the lid of the Petri dishes. Guanidine was extracted from rosettes in 80% (v/v) methanol containing $^{13}C^{15}N$-labeled guanidine as analytical standard and quantified by LC-MS. *din11-1*, single T-DNA insertion mutant of *Din11*. The yellow line marks a change in the scale of the Y-axis. Error bars indicate SD of four independent plant samples. The *din11-2* mutant line and the independently generated second triple mutant gave consistent results.

The online version of this article includes the following source data and figure supplement(s) for figure 6:

**Source data 1.**

**Figure supplement 1.** 2-Aminoadipate content after homoarginine feeding.

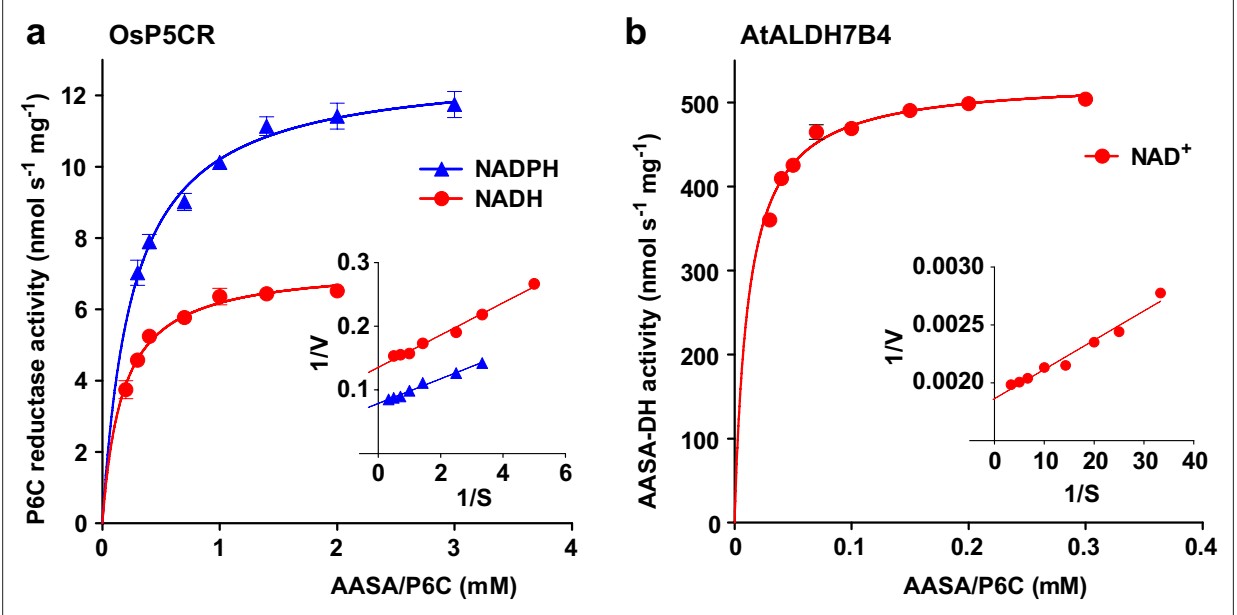

**Figure 7.** Reduction and oxidation of AASA/P6C by plant enzymes. Recombinant purified enzymes were used to compare the reduction of P6C to pipecolate by pyrroline-5-carboxylate reductase from rice (OsP5CR; **a**) with the oxidation of AASA by Arabidopsis aldehyde dehydrogenase 7B4 (ALDH7B4; **b**) at 35 °C with a fixed cofactor concentration of 1 mM. OsP5CR used either NADH or NADPH as electron donor for the conversion of P6C to pipecolate. ALDH7B4 had a much higher specific activity and a lower $K_M$, but efficiently catalyzed the reaction of AASA to aminoadipate only with NAD[+] as electron acceptor. Data points are the average ± SD of technical triplicates. Colored lines represent the fit of the experimental data to the Michaelis-Menten equation. The insets show the Lineweaver-Burk plots of the same data. All experiments were repeated with an independent enzyme preparation and showed consistent results.

The online version of this article includes the following source data and figure supplement(s) for figure 7:

**Source data 1.**

**Figure supplement 1.** Characterization of ALDH7B4 (At1g54100).

**Figure supplement 1—source data 1.**

further been reported that expression of Din11 was induced by treatment with methyljasmonate (MeJA; *Caarls et al., 2017*) and by perturbations of ROS homeostasis (*Gadjev et al., 2006*), whereas analysis of publicly available RNAseq data did not reveal a treatment that induced all three Arabidopsis 2-ODD-C23 isoforms (*Figure 1—figure supplement 3*; *Zhang et al., 2020*). Treatment of WT plants with 10 or 50 µM MeJA did not induce consistent and marked changes of the content of homoarginine and arginine in the rosettes of Arabidospis seedlings. However, MeJA treatment strongly induced the guanidine content in Arabidopsis rosettes and this induction was lost in *din11* single mutants and not decreased much further in 2-ODD-C23 triple mutants (*Figure 6b*).

## Metabolic fate of AASA in plants

The different rates of homoarginine metabolism in WT plants, triple mutants and At3g50210:GFP overexpressing plants, as evidenced by different levels of guanidine accumulation, were surprisingly not reflected in different rates of homoarginine accumulation after external supply (*Figure 4b*). Therefore, we wondered if AASA/P6C production by 2-ODD-C23 might serve a signaling rather than a metabolic function. As in the *E. coli* cultures, P5CR might convert AASA/P6C to pipecolate, which is the direct precursor for the mobile defense signal N-hydroxypipecolate (*Zeier, 2021*). Purified, recombinant plant P5CR converted P6C to pipecolate with a $K_M$ of 173±18 µM or 257±23 µM and an $A_{max}$ of 7 or 13 nmol s$^{-1}$ mg$^{-1}$, when NADH or NADPH were used as electron donors, respectively (*Figure 7a*). An alternative route for AASA/P6C metabolism would be oxidation of AASA to aminoadipate by an aldehyde dehydrogenase. For maize and pea, it was shown that ALDH7B has a high affinity for AASA (*Končitíková et al., 2015*). Indeed, purified, recombinant AtALDH7B4 (At1g54100) converted AASA to aminoadipate with a $K_M$ of 11.9±1.0 µM and an $A_{max}$ of 529±6 nmol s$^{-1}$ mg$^{-1}$ with NAD[+] as electron acceptor (*Figure 7b*). For the cofactor NAD[+], ALDH7B4 had a $K_M$ of 324±17 µM, whereas the enzyme

was almost inactive with NADP$^+$ as electron acceptor and neither was NADPH formation linear over time nor did it show a linear correlation with the amount of enzyme (*Figure 7—figure supplement 1a and b*). ALDH7B4 also accepted GSA/P5C as substrate, but the conversion rates were 60 times lower compared to AASA/P6C and also in this case, the reaction rate was not linearly correlated to the amount of enzyme (*Figure 7—figure supplement 1*). The 10-fold lower $K_M$ of AASA oxidation to aminoadipate *vs.* P6C reduction to pipecolate suggests that the oxidative route is predominantly occurring under physiological conditions. Independent of genotype and homoarginine supply, the pipecolate content of seedlings was below 2 nmol (g Fw)$^{-1}$ under our conditions and could not be quantified because the peak was not clearly separated from other compounds with the same M/Z ratio. The aminoadipate content in Arabidopsis rosettes was also below the level for reliable quantification [1 nmol (g Fw)$^{-1}$] but increased to almost 6 nmol (g Fw)$^{-1}$ in At3g50210:GFP-overexpression plants after supply of external homoarginine (*Figure 6—figure supplement 1*). This increase was absent in 2-ODD-C23 triple mutants and barely detectable in WT plants, indicating that aminoadipate metabolism is faster than the production via AASA derived from homoarginine hydroxylation by 2-ODD-C23.

## Discussion

In this study, we discovered a novel enzymatic activity for a previously uncharacterized clade of plant 2-ODDs. All analyzed 2-ODD-C23 proteins mediated the hydroxylation of homoarginine at the C6 position, thereby inducing its decay to AASA and guanidine (*Figure 3*). This is in sharp contrast to *P. savastanoi* EFE, which was virtually inactive with homoarginine (*Martinez and Hausinger, 2016*). The $K_M$ values for homoarginine hydroxylation by the plant enzymes were higher than the $K_M$ of *P. savastanoi* EFE for arginine (50 µM) and the $A_{max}$ values were intermediate between the values for ethylene formation (54 nmol s$^{-1}$ mg$^{-1}$) and arginine hydroxylation (1.2 nmol s$^{-1}$ mg$^{-1}$) reported for EFE (*Martinez and Hausinger, 2016*). Among the biochemically characterized plant 2-ODDs from class C, DMR6/S5H had a lower $K_M$ (5.1±1.4 µM) and a similar $A_{max}$ (5.9±1.1 nmol s$^{-1}$ mg$^{-1}$) for salicylate hydroxylation (*Zhang et al., 2017b*). Din11, which is exclusively present in *Brassicaceae*, additionally accepted arginine as substrate. Although Din11 has a lower specific activity and a higher $K_M$ for the reaction with arginine, the low concentration of homoarginine in Arabidopsis may lead to a preferential hydroxylation of arginine by Din11. 2-ODD33, the 2-ODD-C23 ortholog from rice, was previously reported to have melatonin-2-hydroxylase activity (*Byeon and Back, 2015*). However, the reported rate of 2-hydroxymelatonin accumulation by purified, recombinant 2-ODD33 was 1000-fold lower than the hydroxylation of homoarginine in our experiments, indicating that homoarginine is the preferred substrate of 2-ODD-C23 enzymes.

The evolutionary origin of 2-ODD-C23 function is at present ambiguous. Loss of the gene in *Chlamydomonas reinhardtii* has previously led to the interpretation that these enzymes were specific to plants (*Kawai et al., 2014*). However, putative orthologs of 2-ODD-C23 are present in other green algae, whereas the protein with the highest degree of similarity in the cyanobacterium *Anabaena variabilis* (Ava5009) did not have homoarginine-6-hydroxylase activity when expressed in *E. coli* (*Figure 1*, *Figure 2*). It is thus conceivable that the production of guanidine from homoarginine or arginine evolved in photosynthetic eukaryotes. The single-copy 2-ODD-C23 isoforms from rice and soybean accepted exclusively homoarginine as substrate, indicating that arginine hydroxylation by Arabidopsis Din11 derived from neofunctionalization after at least two duplication events in the *Brassicaceae* family (*Figure 1*). Sequence alignments between bacterial and fungal EFEs and plant 2-ODDs did not reveal a close relationship between Din11 and EFE (*Figure 1—figure supplement 1*), indicating an independent evolution of arginine-5-hydroxylase activity in both enzymes.

The guanidine content was strongly reduced in mutant plants devoid of 2-ODD-C23 expression, indicating that 2-ODD-C23s are the major source of free guanidine in Arabidopsis. At3g50210 mediated the major part of the homoarginine-dependent guanidine production (*Figure 5—figure supplement 1*) which is consistent with the highest specific activity (*Figure 3*) and the highest basal expression level of At3g50210 (*Figure 1—figure supplement 3*). In contrast, Din11 expression has the highest dynamic range and induction of Din11 by MeJA caused a strong increase in guanidine production, presumably from arginine as precursor (*Figure 6*). This increase was virtually absent in *din11* single mutants and the triple mutants, indicating that it was exclusively attributable to Din11 activity. Based on a basal guanidine content of 120 nmol (g dry weight)$^{-1}$ (equivalent to 7 ppm) in

Arabidopsis and our observation that other plants have similar or even higher contents of guanidine (*Figure 5—figure supplement 2*), we estimated the global annual production of guanidine by plants. With a total annual biomass production of $5*10^{10}$ t by land plants, this estimation yields an annual guanidine production of $3.5*10^5$ t by plants, corresponding roughly to the 2.5-fold of the industrial production in 2016 (*Precision Reports, 2021*).

Plants were shown to transport but barely metabolize guanidine (*Marsden et al., 2015*), indicating that nitrogen released as guanidine cannot be efficiently recycled by plants. Considering the often growth-limiting function of nitrogen for plants, this raises the question whether guanidine may serve as a defense or signaling compound. In soil, a greater proportion of bacteria contains genes for guanidine import and assimilation than among bacteria associated with human skin or the gut, which preferentially contain genes for guanidine exporters (*Sinn et al., 2021*). Since bacteria differ in their responses to guanidine, it is possible that guanidine accumulation or even secretion is a way for plants to modulate their microbiomes (*Sharma et al., 2023*). A similar function was proposed for hydroxyguanidine that was secreted by *Pseudomonas canavaninivorans* after degradation of plant-produced canavanine by a PLP-dependent canavanine-γ-lyase (*Hauth et al., 2022*). Induction of Din11 expression and guanidine production by wounding and MeJA may also point to a function of guanidine in the defense against or repulsion of herbivores (*Caarls et al., 2017*; *Griffiths, 2020*). Expression of Din11 was additionally induced in several mutants or after treatments that cause ROS accumulation, indicating that ROS may act as second messenger in the induction of Din11 expression (*Figure 1—figure supplement 3*; *Gadjev et al., 2006*). Further studies will be required to determine the biological purpose of guanidine production from homoarginine or arginine by 2-ODD-C23 enzymes.

In addition or alternatively, the second reaction products of 2-ODD-C23s, AASA/P6C or GSA/P5C could represent physiologically functional products. GSA/P5C produced by cytosolic Din11 (*Figure 4*) is most likely converted to proline by P5CR, which is a cytosolic enzyme in plants as well (*Funck et al., 2012*). Thereby, the activity of Din11 may explain the observation that besides glutamate also arginine can serve as precursor for proline synthesis in plants (*Adams and Frank, 1980*; *Winter et al., 2015*). So far, no combination of enzymes residing in the same compartment was known that could catalyze the conversion of arginine to proline without glutamate as intermediate (*Trovato et al., 2019*). However, the exclusive occurrence of Din11 orthologs in *Brassicaceae* indicates that this metabolic pathway is not common among plants (*Figure 1*). The maximal guanidine concentration of 400 nmol (g Fw)$^{-1}$ observed in MeJA-treated Arabidopsis plants (*Figure 6*) argues against a major contribution to stress-induced proline accumulation, which typically reaches 5–10 μmol (g Fw)$^{-1}$ (*Forlani et al., 2019*). Besides the induction by MeJA, expression of Din11 was also induced by prolonged dark treatment, which led to its initial identification and naming (*Fujiki et al., 2001*). Prolonged dark treatment causes C-starvation in plants and results in the induction of autophagy to mobilize energy by breakdown of cellular components and organelles (*Magen et al., 2022*). Degradation of homoarginine by 2-ODD-C23s is unlikely to be a relevant contribution to energy production during C-starvation because the observed homoarginine content was very low in Arabidopsis (*Figure 5*). In contrast, degradation of arginine via Din11 to GSA/P5C and guanidine may serve to mobilize the C-skeleton of arginine without overloading the nitrogen metabolism. Degradation of arginine via arginase to ornithine and urea would result ultimately in glutamate and ammonia formation by glutamate dehydrogenase and urease (*Winter et al., 2015*). High ammonia concentrations are toxic for plants, especially when the energy or the enzymes to re-assimilate it into glutamine are lacking (*Moreira et al., 2022*). However, the extent of chlorophyll degradation was not different between WT plants, 2-ODD-C23 triple mutants and overexpressing plants, indicating that the efficiency of autophagy in remobilizing carbon and nitrogen does not depend on 2-ODD-C23 activity (*Figure 6*). We are not aware of any reports describing a function of homoarginine in plants except the storage and defense function in seeds of hyperaccumulating *Lathyrus* species (*Bell, 1980*). As we were not able to resolve homoarginine from methylarginine in our plant extracts, we can at present not interpret or explain the reduction of the content of the substance with m/z ratio 189 in both, 2-ODD-C23 overexpression plants and 2-ODD-C23 triple mutants.

AASA originating from the decay of 6-hydroxyhomoarginine can be converted to pipecolate by P5CR after spontaneous cyclisation to P6C (*Fujii et al., 2002*; *Struys et al., 2014*). In this way, 2-ODD-C23s might contribute to the production of the mobile defense signal N-hydroxypipecolate (*Zeier, 2021*). However, the direct comparison of the kinetic properties of P5CR and ALDH7B4

suggested that oxidation of AASA to aminoadipate by ALDH7B4, which is a shared reaction with the lysine degradation pathway, is the main pathway under physiological conditions (*Figure 7*; *Arruda and Barreto, 2020*). We did not find information about the rate of AASA-cyclisation to P6C or the equilibrium concentrations in plants. In human plasma from patients with ALDH7A1 defects, approximately equal concentrations of both compounds were detected after FMOC-derivatization (*Xue et al., 2019*). Consistent with the kinetic properties of P5CR and ALDH7B4, no increase of the pipecolate content after feeding with external homoarginine was observed in Arabidopsis plants overexpressing the 2-ODD-C23 isoform At3g50210. Instead, a small increase in aminoadipate content was observed, indicating that AASA was indeed oxidized by ALDH7B4, but the rate of aminoadipate metabolism apparently exceeded the rate of synthesis by 2-ODD-C23 and ALDH7B4 (*Figure 6—figure supplement 1*). *E. coli* lacks a specific AASA-DH and consequently cyclisation to P6C followed by reduction to pipecolate by the constitutively expressed P5CR (proC) is the main route of homoarginine metabolism in 2-ODD-C23 expressing *E. coli* cells (*Figure 2*).

In humans, homoarginine is formed from lysine and arginine by mitochondrial glycine amidinotransferase (*Choe et al., 2013*). It is discussed that homoarginine interferes with NO production from arginine and thus might be an important regulator of NO signaling (*Koch et al., 2023*). Plants seem to lack NO synthases that use arginine as substrate and do not have proteins with significant sequence similarity to glycine amidinotransferase, although a protein with similar activity has been isolated from *Lathyrus sativus* seedlings (*León and Costa-Broseta, 2020*; *Srivenugopal and Adiga, 1980*). Thus, the biosynthesis and physiological function of homoarginine in plants will need to be addressed in future studies. In humans, the accumulation of AASA as consequence of mutations in ALDH7A1/Antiquitin, which is the ortholog of Arabidopsis ALDH7B4, leads to developmental retardation and epilepsy (*Bouchereau and Schiff, 2020*; *Brocker et al., 2013*). The disease symptoms are caused by formation of adducts of AASA with PLP, thereby inactivating PLP-dependent enzymes. In plants, ALDH7B enzymes may similarly serve to detoxify AASA derived from homoarginine hydroxylation by 2-ODD-C23s or from lysine degradation by the bifunctional lysine-ketoglutarate reductase/saccharopine dehydrogenase (LKR/SDH, At4g33150) (*Arruda et al., 2000*; *Goncalves-Butruille et al., 1996*). Mutation of Arabidopsis ALDH7B4 was reported to increase drought and salt sensitivity, whereas the sensitivity to herbicides was not significantly altered (*Gil-Monreal et al., 2017*; *Kotchoni et al., 2006*). In rice, mutation of ALDH7B6 reduced seed viability and caused the accumulation of pigments in the endosperm (*Shen et al., 2012*; *Shin et al., 2009*). Our analyses show that Arabidopsis ALDH7B4 has an even lower $K_M$ (11.9 µM, *Figure 7*) for AASA than the previously analyzed orthologs from rice and pea (97 and 89 µM respectively; *Končitíková et al., 2015*). It remains to be determined if ALDH7B4 functions specifically to prevent AASA accumulation or if it additionally acts nonspecifically on a broad range of aldehydes and semialdehydes.

Overall, our characterization of plant 2-ODD-C23s as homoarginine-6-hydroxylases is to our knowledge the first description of eukaryotic enzymes that mediate the production of guanidine. Although homoarginine represents a low abundant metabolite of unknown function and origin in most analyzed plants, the large amount of plant biomass produced every year implies that substantial amounts of guanidine enter the biological nitrogen cycle via the action of 2-ODD-C23 enzymes. The second product of 6-hydroxyhomoarginine decay, AASA/P6C, was primarily fed into the lysine degradation pathway by ALDH7B. So far, Din11 from Arabidopsis was the only 2-ODD-C23 isoform that accepted also the abundant arginine as substrate, but its low expression level and low specific activity apparently prevented excessive diversion of arginine toward guanidine and proline production. Consistently, the reduced guanidine content was so far the only phenotypical difference that we detected in Arabidopsis mutants lacking all three 2-ODD-C23 isoforms. Although neither homoarginine nor guanidine are highly abundant metabolites in Arabidopsis, the observed differences in the triple mutants and overexpressor plants confirm that the 2-ODD-C23 isoforms can catalyze homoarginine hydroxylation *in vivo*. Induction of Din11 by prolonged dark treatment, wounding, and reactive oxygen species provides hints that guanidine, AASA, or GSA may function in defense responses or other biotic interactions. We speculate that plants use guanidine production to influence guanidine-sensing or guanidine-assimilating microbes and plan to test this hypothesis in future studies.

## Materials and methods

**Key resources table**

| Reagent type (species) or resource | Designation | Source or reference | Identifiers | Additional information |
|---|---|---|---|---|
| Gene (*Arabidopsis thaliana*) | Din11 (At3g49620); At3g49630; At3g50210; P5CR (AT5G14800); ALDH7B4 (AT1G54100) | TAIR | dito | |
| Gene (*Oryza sativa*) | ODD33 (Os09g07020) | GenBank | NM_001422704 | |
| Gene (*Glycine max*) | LOC100789278 | GenBank | NM_001255066 | |
| Gene (*Anabaena variabilis*) | Ava5009 | Genbank | CP000117 | |
| Strain/background (*Arabidopsis thaliana*) | Col-0 wildtype | NASC | N60000 | |
| Strain/background (*E. coli*) | SoluBL21 | Amsbio (Genlantis) | AMS.C700200 | |
| Genetic reagent (*A. thaliana* T-DNA insertion lines) | see **Supplementary file 2** | NASC | | see **Supplementary file 2** |
| Antibody | Anti-GFP, (rat, monoclonal) | ChromoTek | 3H9, RRID: AB_10773374 | (1:1000) |
| Antibody | Anti-rat HRP (goat, polyclonal) | Roche | A9037 | (1:80,000) |
| Recombinant DNA reagent | pET24a | Novagen (Merck) | 69749 | |
| Recombinant DNA reagent | pET24-HisTEV | this study | | Data **Source data 2** |
| Recombinant DNA reagent | pENTRa1 | Invitrogen (ThermoFisher) | A10462 | |
| Recombinant DNA reagent | pUBC-GFP-Dest | **Grefen et al., 2010**; doi:10.1111/j.1365–313X.2010.04322.x | | |
| Recombinant DNA reagent | pHEE401E | **Wang et al., 2015**; doi: 10.1186/s13059-015-0715-0 | | |
| Sequence-based reagent | custom DNA oligos | Sigma-Aldrich (Merck) | | see **Supplementary file 3** |
| Peptide, recombinant protein | see 'genes' above | This study | | see 'Materials and methods' |
| Commercial assay or kit | NEBuilder HiFi | NEB | E2621L | |
| Chemical compound, drug | D,L-P5C, L-P6C | This study | | see 'Materials and methods' |
| Software, algorithm | Prism V.5 | GraphPad | | |
| Software, algorithm | Excel 2016 | Microsoft | | |
| Software, algorithm | ZEN | Carl Zeiss Microscopy | | |
| Software, algorithm | Fiji/ImageJ | Fiji/NIH | | |
| Software, algorithm | OxyTrace+ | Hansatech | | |
| Software, algorithm | LabSolutions LC/GC, V5.93 | Shimadzu | | |

### DNA constructs and recombinant protein production

Bacterial expression constructs for Arabidopsis Din11, At3g49620, At3g50210, and ALDH7B4, as well as Os2-ODD33 and Gm2-ODD-C23 were generated by amplifying the coding sequence from cDNA with primers (**Supplementary file 3**) that introduce overlaps with a modified pET vector containing an N-terminal His-tag and a TEV cleavage site (**Source data 2**). The resulting PCR products were fused with the linearized vector by Gibson assembly (NEBuilderHiFi). A plasmid for expression of 6xHis-SUMO-tagged *Ps*EFE was obtained from Christopher Schofield (**Zhang et al., 2017a**). Protein expression in *E. coli* SoluBL21 cells was induced with 1 mM IPTG at 18 °C overnight. The cells were lysed by sonication in extraction buffer [50 mM Hepes pH 7.5, 50 mM NaCl, 1 x protease inhibitor

cocktail (EDTA-free cOmplete, Roche)] and the His-tagged proteins were purified with HisBind NiNTA-agarose (QIAGEN) according to the recommendations of the supplier. Imidazole was removed from the purified proteins by passage through PD MidiTrap G-25 columns (GE lifescience) equilibrated with extraction buffer. Protein concentrations were determined according to *Bradford, 1976* with BSA as standard. All activity assays were performed with enzymes purified from at least three independent bacterial cultures.

## Enzyme activity assays

Activity of ODDs was measured by following the oxygen consumption at 30 °C in a Clark-type electrode (Hansatech Oxygraph+) calibrated according to the instructions of the supplier. The reaction mix contained 50 mM MOPS pH 7.2, 70 mM NaCl, 20 mM KCl, 0.4 mM Na-ascorbate, 40 µM Fe(N-H$_4$)$_2$(SO$_4$)$_2$, and variable concentrations of substrate or cosubstrate. The reactions were started by adding the purified enzymes and the specific activities were calculated from the difference between the oxygen consumption rate in the initial linear phase and the rate observed prior to the addition of the enzymes (typically around –20 nmol O$_2$ ml$^{-1}$ s$^{-1}$, corresponding to 1 to 2% of the maximal rate in the presence of enzyme).

The activity of P5CR was calculated from the difference in the rates of NAD(P)H oxidation at 35 °C in the presence or absence of DL-P5C or L-P6C as described previously (*Forlani et al., 2015*). The standard enzyme assay contained 50 mM Tris-HCl, pH 7.5, 0.5 mg ml$^{-1}$ NAD(P)H and 2 mM DL-P5C or 1 mM L-P6C. DL-P5C was synthesized by the periodate oxidation of δ-allo-hydroxylysine (Sigma H0377) and purified by cation-exchange chromatography, as described previously (*Forlani et al., 1997*). L-P6C generation from L-allysine ethyleneacetal (Sigma-Aldrich 714208) roughly followed a previous protocol (*de La Fuente et al., 1997*). A solution of 50 mg allysine ethyleneacetal in 5 ml water was mixed with 300 mg amberlite IR120 (Fluka 06428) and gently stirred for 20 min. The supernatant was discarded and the resin was loaded into a mini-column, washed with 10 mL of water and eluted with 10 mL of 1 M HCl in 1 ml fractions. The concentration of P6C was measured by mixing a proper dilution in 60 µl with 120 µl 1 mg ml$^{-1}$ o-aminobenzaldehyde in ethanol, and the increase of absorbance at 465 nm was followed until a plateau was reached. The extinction coefficient of ε$_{465}$=2,800 M$^{-1}$ cm$^{-1}$ was used (*Fothergill and Guest, 1977*). Both DL-P5C and L-P6C were kept as stocks in 1 M HCl at 4 °C in the dark and neutralized with 1 M Tris-base immediately before use. The activity of ALDH7B4 was calculated from the difference in NAD(P)$^+$ reduction at pH 7.5 and 35 °C in the presence or absence of variable concentrations of DL-P5C or L-P6C. The standard reaction mixture contained 50 mM Tris-HCl, pH 7.5, 1 mM NAD(P)$^+$ and 2 mM DL-P5C or 1 mM L-P6C. The extinction coefficient of ε$_{340}$=6.22 mM$^{-1}$ cm$^{-1}$ for NADH and NADPH was used to calculate specific activities.

## Plant material, plant transformation, and cultivation conditions

Arabidopsis [*Arabidopsis thaliana* (L.) Heynh.], ecotype Col-0 and T-DNA insertion lines (*Supplementary file 2*) were obtained from the NASC. Surface-sterilized seeds were sown on solid half-strength Murashige and Skoog medium including Gamborg B5 vitamins (pH 5.8, supplemented with 2% [w/v] sucrose and 0.8% agar) in Petri dishes sealed with Leukopor tape (Leukoplast) and cultivated under a long-day light regime (16 h photoperiod; 110 µmol m$^{-2}$ s$^{-1}$ at 21 °C). For seed production, seedlings were transferred to commercial substrate ('Einheitserde', type P, Gebr. Patzer, Sinntal-Altengronau, Germany) and cultivated in a greenhouse under long-day conditions. All T-DNA insertion mutants were backcrossed at least three times with WT plants, until the segregation pattern was consistent with a single T-DNA insertion. Constructs for the expression of GFP fusion proteins were obtained by inserting the coding sequences into pENTRa1 (Invitrogen) followed by LR-recombination with pUBC-GFP-Dest (*Grefen et al., 2010*). The construct for simultaneous CRISPR/Cas9-mediated inactivation of *Din11* and *At3g49630* was generated by ligating two matching oligonucleotides (*Supplementary file 3*) into the BsaI-linearized vector pHEE401E (*Wang et al., 2015*). The resulting gRNA matches a Cas9 cleavage site in both genes with one mismatch in Din11 and no strongly predicted off-targets. All constructs were introduced into Arabidopsis plants by floral dip with *Agrobacterium tumefaciens* strain GV3101 (*Clough and Bent, 1998*). Transgenic plants were selected by spraying of seedlings with BASTA solution (50 µg ml$^{-1}$) or growth on sterile culture plates containing 30 µg ml$^{-1}$ hygromycin B (Sigma). For the GFP constructs, lines with stable expression and a single-copy T-DNA insertion were selected by segregation analysis. CRISPR/Cas9-induced mutations were detected by high-resolution

melting analysis using gene-specific primers (*Supplementary file 3*) and confirmed by sequencing of the PCR products. The CRISPR/Cas9 construct was eliminated by backcrossing to WT plants. The absence of functional transcripts was confirmed by RT-PCR and sequencing of the products.

For feeding experiments, homoarginine or arginine were added to the culture medium from filter-sterilized stocks (100 mM, pH 6). For treatment with MeJA, 100 µl of 5 mM MeJA in 50% (v/v) EtOH were distributed on the lid of the Petri dishes, which were consecutively sealed with Parafilm. Control plates were prepared with 50% EtOH. For the analysis of starvation and autophagy, the plants were cultivated on half-strength MS plates without sucrose under short-day conditions at 21 °C. Plates with 8-day-old seedlings were wrapped in aluminum foil and incubated at 21 °C for 8 days.

Other plant material was collected around the University of Konstanz or was purchased in a garden store. Seeds of alfalfa, broccoli, and rocket were surface sterilized as above and germinated on filter discs soaked in distilled water.

## Confocal microscopy

Protoplasts were isolated from leaves of 2-ODD-C23:GFP expressing and WT plants by overnight incubation in protoplast medium (0.45 M sorbitol, ½ strength MS salt mixture) supplemented with 10 mg ml$^{-1}$ cellulase and 2.5 mg ml$^{-1}$ macerozyme. Spectral images were recorded with a Zeiss LSM880 with a 63 x water immersion lens. The samples were excited at 488 nm and images were recorded with 20 spectral channels (9 nm bandwidth) between 490 and 668 nm. A chlorophyll spectrum was obtained from WT protoplasts and a GFP spectrum from an epidermis protoplast that did not contain chloroplasts. Linear unmixing with background subtraction was performed with the ZEN software (Zeiss). Channel overlay, false coloring and adjustment to identical gain, offset and contrast settings were performed in ImageJ and Adobe Photoshop.

## Metabolite analyses

*E. coli* cultures were lysed by ultrasonication prior to or after the addition of a 20-fold excess of MeOH spiked with 10 µM $^{13}C^{15}N$-guanidine (Sigma-Aldrich 607312). Plant samples were homogenized at –80 °C in a TissueLyser (Qiagen) and resupendend in a 10-fold excess of 80% (w/v) MeOH containing 10 µM $^{13}C^{15}N$-guanidine and 20 µM $^{13}C^{15}N$-arginine (Cortectnet CCN250). After ultrasonication for 5 min, the suspension was centrifuged for 10 min at maximum speed in a tabletop centrifuge and the supernatant was used directly for metabolite quantification.

Metabolite concentrations in the samples were determined by high performance liquid chromatography coupled with mass spectrometry on a LC-MS-2020 Single Quadrupole mass spectrometer (Shimadzu). Two µL sample volume was injected into a Nucleodur HILIC column (250 mm * 2 mm, 3 µm particle size, Macherey-Nagel). Metabolites were eluted at 20 °C with an acetonitrile gradient (90% (v/v) to 60% during 6 min, to 45% in 1 min, constant for 2.5 min, back to 90% in 1.5 min) in 10 mM ammonium formate, 0.2% (v/v) formic acid at a flow rate of 0.15 ml min$^{-1}$. The separated compounds were analyzed by single ion monitoring (SIM) for positively (and negatively when appropriate) charged ions. Quantification of guanidine, pipecolate and aminoadipate was performed by calibrating peak areas with pure standards and by normalization to the added $^{13}C^{15}N$-guanidine. Homoarginine was quantified by calibrating the peak areas with a pure standard and by normalization to the added $^{13}C^{15}N$-arginine.

Ethylene production by *E. coli* cultures was analyzed with an SRI flame ionization detector (FID) equipped with a HayeSep-D GC column. Serial dilutions of pure ethylene were use to calibrate the FID signal.

## Material availability

All the genetic material generated for this study is available from the corresponding authors upon reasonable request.

## Acknowledgements

JSH acknowledges the ERC CoG project 681777 "RiboDisc" for financial support. We thank Astrid Joachimi, Dmitry Galetskiy, and Janine Dietrich for technical assistance as well as Nicolai Müller and Dieter Spiteller for help with metabolite detection. Additionally, we thank Svetlana Boycheva Woltering for help with autophagy assays as well as Martin Stöckl and the Bioimaging Centre of the

University of Konstanz for assistance with confocal microscopy and image processing. We thank Christopher Schofield for providing the *Ps*EFE expression construct. We are grateful to Erika Isono and the team of gardeners of the University of Konstanz for access to plant cultivation facilities and plant care.

## Additional information

### Funding

| Funder | Grant reference number | Author |
|---|---|---|
| European Research Council | 681777 | Jörg S Hartig |

The funders had no role in study design, data collection and interpretation, or the decision to submit the work for publication.

### Author contributions

Dietmar Funck, Conceptualization, Resources, Data curation, Formal analysis, Investigation, Visualization, Writing – original draft, Writing – review and editing; Malte Sinn, Conceptualization, Resources, Data curation, Formal analysis, Investigation, Methodology, Writing – review and editing; Giuseppe Forlani, Resources, Data curation, Formal analysis, Investigation, Visualization, Methodology, Writing – review and editing; Jörg S Hartig, Conceptualization, Resources, Supervision, Funding acquisition, Writing – original draft, Project administration, Writing – review and editing

### Author ORCIDs

Dietmar Funck ⓘ http://orcid.org/0000-0002-9855-0419
Malte Sinn ⓘ http://orcid.org/0000-0002-3790-9686
Giuseppe Forlani ⓘ http://orcid.org/0000-0003-2598-5718

Reviewer #1 (Public Review): https://doi.org/10.7554/eLife.91458.3.sa1
Reviewer #2 (Public Review): https://doi.org/10.7554/eLife.91458.3.sa2
Author response https://doi.org/10.7554/eLife.91458.3.sa3

## Additional files

### Supplementary files

• Supplementary file 1. Kinetic constants of Arabidopsis 2-ODD-C23 enzymes.

• Supplementary file 2. 27Arabidopsis T-DNA insertion lines used in this study.

• Supplementary file 3. Sequences of primers used in this study. Supplementary file 3 references: *Alonso et al., 2003Kleinboelting et al., 2012Sessions et al., 2002*

• MDAR checklist

• Source data 1. Source data relating to *Supplementary file 1*.

• Source data 2. Annotated sequence of modified pET24a vector.

### Data availability

All data pertaining to this article is presented within the article, its supplements and source data files.

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
