## [Editor Report · eLife assessment]

This **fundamental** study advances our understanding of nitrogen metabolism by identifying a new type of guanidine-forming enzyme in eukaryotes. The key claims of the article are **convincingly** supported by the data, with meticulous biochemical, cellular, and *in vivo* studies on guanidine production. The work will stimulate interest in the cellular roles of homoarginine, and, more generally, in the biochemistry and metabolism of guanidine derivatives.

---

## [Referee Report · Reviewer #1 (Public Review)]

Nitrogen metabolism is of fundamental importance to biology. However, the metabolism and biochemistry of guanidine and guanidine containing compounds, including arginine and homoarginine, have been understudied over the last few decades. Very few guanidine forming enzymes have been identified. Funck et al define a new type of guanidine forming enzyme. It was previously known that 2-oxogluturate oxygenase catalysis in bacteria can produce guanidine via oxidation of arginine. Interestingly, the same reported enzyme that produces guanidine from arginine also oxidises 2-oxogluturate to give the plant signalling molecule ethylene. Funck et al show that a mechanistically related oxygenase enzyme from plants can also produce guanidine, but instead of using arginine as a substrate, it uses homoarginine and does not produce ethylene. The work will stimulate interest in the cellular roles of homoarginine, a metabolite present in plants and other organisms including humans and, more generally, in the biochemistry and metabolism of guanidine derivatives.

1. Significance

Studies on the metabolism and biochemistry of the small nitrogen rich molecule guanidine and related compounds including arginine have been largely ignored over the last few decades. Very few guanidine forming enzymes have been identified. Funck et al define a new guanidine forming enzyme that works by oxidation of homoarginine, a metabolite present in organisms ranging from plants to humans. The new enzyme requires oxygen and 2-oxogluturate as cosubstrates and is related, but distinct from a known enzyme that oxidises arginine to produce guanidine, but which can also oxidise 2-oxogluturate to produce the plant signalling molecule ethylene.

I thought this was an exceptionally well-written and interesting manuscript. Although a 2-oxogluturate dependent guanidine forming enzyme is known (EFE), the discovery that a related enzyme oxidises homoarginine is really interesting, especially given the presence of homoarginine in plant seeds. There is more work to be done in terms of functional assignment, but this can be the subject of future studies. I also fully endorse the authors' view that guanidine and related compounds have been massively understudied in recent times. Congratulations to the authors on a very nice study.

Overall, I thought this was a very interesting study, comprising biochemical, cellular, and in vivo studies. Of course, more could be done on each of these, and likely will be, but I think the assignment of biochemical function is very strong, across all three approaches. The one new experiment I requested was a demonstration of whether ethylene is produced by the new enzymes - this was clearly shown not to be the case.

---

## [Referee Report · Reviewer #2 (Public Review)]

In this study, Dietmar Funck and colleagues have made a significant breakthrough by identifying three isoforms of plant 2-oxoglutarate-dependent dioxygenases (2-ODD-C23) as homo/arginine-6-hydroxylases, catalyzing the degradation of 6-hydroxyhomoarginine into 2-aminoadipate-6-semialdehyde (AASA) and guanidine. This discovery marks the very first confirmation of plant or eukaryotic enzymes capable of guanidine production.

The authors selected three plant 2-ODD-C23 enzymes with the highest sequence similarity to bacterial guanidine-producing (EFE) enzymes. They proceeded to clone and express the recombinant enzymes in E coli, demonstrating capacity of all three Arabidopsis isoforms to produce guanidine. Additionally, by precise biochemical experiments, the authors established these three 2-ODD-C23 enzymes as homoarginine-6-hydroxylases (and arginine-hydroxylase for one of them). Furthermore, the authors utilized transgenic plants expressing GFP fusion proteins to show the cytoplasmic localization of all three 2-ODD-C23 enzymes. Most notably, using T-DNA mutant lines and CRISPR/Cas9-generated lines, along with combinations of them, they demonstrate the guanidine-producing capacity of each enzyme isoform in planta. These results provide robust evidence that these three 2-ODD-C23 Arabidopsis isoforms are indeed homoarginine-6-hydroxylases responsible for guanidine generation.

The findings presented in this manuscript are a significant contribution for our understanding of plant biology, particularly given that this work is the first demonstration of enzymatic guanidine production in eukaryotic cells. However, there are a couple of concerns and potential ways for further investigation that the authors should (consider) incorporate.

Firstly, the observation of cytoplasmic and nuclear GFP signals in the transgenic plants may also indicate cleaved GFP from the fusion proteins. Thus, the authors should perform a Western blot analysis to confirm the correct size of the 2-ODD-C23 fusion proteins in the transgenic protoplasts.

Secondly, it may be worth measuring pipecolate (and proline?) levels under biotic stress conditions (particularly those that induce transcript changes of these enzymes, Fig S8). Given the results suggesting a potential regulation of the pathway by biotic stress conditions (eg. meJA), these experiments could provide valuable insights into the physiological role of guanidine-producing enzymes in plants. This additional analysis may give a significance of these enzymes in plant defense mechanisms.

---

## [Author Response]

The following is the authors’ response to the original reviews.

**Public Reviews:**

**Reviewer #1 (Public Review):**
Nitrogen metabolism is of fundamental importance to biology. However, the metabolism and biochemistry of guanidine and guanidine containing compounds, including arginine and homoarginine, have been understudied over the last few decades. Very few guanidine forming enzymes have been identified. Funck et al define a new type of guanidine forming enzyme. It was previously known that 2-oxogluturate oxygenase catalysis in bacteria can produce guanidine via oxidation of arginine. Interestingly, the same enzyme that produces guanidine from arginine also oxidises 2-oxogluturate to give the plant signalling molecule ethylene. Funck et al show that a mechanistically related oxygenase enzyme from plants can also produce guanidine, but instead of using arginine as a substrate, it uses homoarginine. The work will stimulate interest in the cellular roles of homoarginine, a metabolite present in plants and other organisms including humans and, more generally, in the biochemistry and metabolism of guanidines.1. SignificanceStudies on the metabolism and biochemistry of the small nitrogen rich molecule guanidine and related compounds including arginine have been largely ignored over the last few decades. Very few guanidine forming enzymes have been identified. Funck et al define a new guanidine forming enzyme that works by oxidation of homoarginine, a metabolite present in organisms ranging from plants to humans. The new enzyme requires oxygen and 2oxogluturate as cosubstrates and is related, but distinct from a known enzyme that oxidises arginine to produce guanidine, but which can also oxidise 2-oxogluturate to produce the plant signalling molecule ethylene.Overall, I thought this was an exceptionally well written and interesting manuscript. Although a 2-oxogluturate dependent guanidine forming enzyme is known (EFE), the discovery that a related enzyme oxidises homoarginine is really interesting, especially given the presence of homoarginine in plant seeds. There is more work to be done in terms of functional assignment, but this can be the subject of future studies. I also fully endorse the authors' view that guanidine and related compounds have been massively understudied in recent times. I would like to see the possibility that the new enzyme makes ethylene explored. Congratulations to the authors on a very nice study.

Response: We thank the reviewer for the positive evaluation of our manuscript. In the revised version, we have emphasized more clearly that we found no evidence for ethylene production by the recombinant enzymes. The other suggestions of the reviewer are also considered in the revised version as detailed below.

**Reviewer #2 (Public Review):**
In this study, Dietmar Funck and colleagues have made a significant breakthrough by identifying three isoforms of plant 2-oxoglutarate-dependent dioxygenases (2-ODD-C23) as homo/arginine-6-hydroxylases, catalyzing the degradation of 6-hydroxyhomoarginine into 2aminoadipate-6-semialdehyde (AASA) and guanidine. This discovery marks the very first confirmation of plant or eukaryotic enzymes capable of guanidine production.The authors selected three plant 2-ODD-C23 enzymes with the highest sequence similarity to bacterial guanidine-producing (EFE) enzymes. They proceeded to clone and express the recombinant enzymes in E coli, demonstrating capacity of all three Arabidopsis isoforms to produce guanidine. Additionally, by precise biochemical experiments, the authors established these three 2-ODD-C23 enzymes as homoarginine-6-hydroxylases (and arginine-hydroxylase for one of them). Furthermore, the authors utilized transgenic plants expressing GFP fusion proteins to show the cytoplasmic localization of all three 2-ODD-C23 enzymes. Most notably, using T-DNA mutant lines and CRISPR/Cas9-generated lines, along with combinations of them, they demonstrate the guanidine-producing capacity of each enzyme isoform in planta. These results provide robust evidence that these three 2-ODD-C23 Arabidopsis isoforms are indeed homoarginine-6-hydroxylases responsible for guanidine generation.The findings presented in this manuscript are a significant contribution for our understanding of plant biology, particularly given that this work is the first demonstration of enzymatic guanidine production in eukaryotic cells. However, there are a couple of concerns and potential ways for further investigation that the authors should (consider) incorporate.Firstly, the observation of cytoplasmic and nuclear GFP signals in the transgenic plants may also indicate cleaved GFP from the fusion proteins. Thus, the authors should perform Western blot analysis to confirm the correct size of the 2-ODD-C23 fusion proteins in the transgenic protoplasts.Secondly, it may be worth measuring pipecolate (and proline?) levels under biotic stress conditions (particularly those that induce transcript changes of these enzymes, Fig S8). Given the results suggesting a potential regulation of the pathway by biotic stress conditions (eg. meJA), these experiments could provide valuable insights into the physiological role of guanidine-producing enzymes in plants. This additional analysis may give a significance of these enzymes in plant defense mechanisms.

Response: We thank also reviewer 2 for the positive evaluation and useful suggestions. We performed the proposed GFP Western blot, which indeed indicated the presences of both, fulllength fusion proteins and free GFP, which can explain the partial nuclear localization. We fully agree that further experiments with biotic and abiotic stress will be required to determine the physiological function of the 2-ODD-C23 enzymes. However, the list of potential experiments is long and they are beyond the scope of the present manuscript.

**Reviewer #1 (Recommendations For The Authors):**
Specific pointsOverall, I thought this was a very interesting study, comprising biochemical, cellular, and in vivo studies. Of course more could be done on each of these, and likely will be, but I think the assignment of biochemical function is very strong, across all three approaches. The one new experiment I would like to see is a clear demonstration of whether ethylene is produced - unlikely but should be tested.

We had mentioned our failure to detect ethylene production by the plant enzymes in the previous version and have made it more prominent and reliable by including ethylene production as positive control in the new Figure 2—figure supplement 2.

AbstractDelete 'hitherto overlooked' - this is implicit 'but is more likely' to 'is likely'?

Agreed and modified

IntroductionSecond sentence - what about relevant small molecule primary metabolites including precursors of proteins/nucleic acids.

We modified the sentence accordingly.

Paragraph 2 - maybe also note EFE produces glutamate semi aldehyde, via arginine C-5 oxidation.

Paragraph 2 has been re-phrased according to your suggestion.

Overall, I thought the introduction was exceptionally well written.Perhaps either in the introduction, or later, note there are other 2OG oxygenases that oxidise arginine/arginine derivatives in various ways, e.g. clavaminate synthase/arginine hydroxylases/desaturases.

We added a sentence mentioning the arginine hydroxylases VioC and OrfP to the introduction and included VioC into the sequence comparison in Figure 1—figure supplement 1 to show that these enzymes, as well as NapI, are very different from EFE and the plant hydroxylases.

ResultsParagraph 1 - qualify similarity and refer to/give a structurally informed sequence alignment, including EFE

A new Figure 1—figure supplement 1 was added with sequence identity values and a structurally informed alignment. The text has been modified accordingly.

Paragraph 2 - briefly state method of guanidine analysis

We included a reference to the M&M section and mentioned LC-MS in paragraph 2.

Figure 1 - trivial point - proteins are not expressed/genes are

We have modified the legend to figure 1. However, we would like to point out that terms like“recombinant protein expression” are widely used in the field. A quick search with googleNgram viewer shows that “protein expression” started to appear in the mid-80ies and its use stayed constantly at 1/8th of “gene expression”.

Define errors clearly in all figure legends, clearly defining biological/technical repeatsPage 6 - was the His-tag cleared to ensure no issues with Ni contamination?

We treat individual plants or independent bacterial cultures as biological replicates. Only in the case of enzyme activity assays with NAD(P)H, technical replicates were used and this has been indicated in the legend of figure 6.

Lower case 'p' in pentafluorobenzyl correctedIn Figure 2 make clear the hydroxylated intermediates are not observed

We now use grey color for the intermediates and have put them in brackets. Additionally we state in the figure legend that these intermediates were not detected.

Pages 6-7 - I may have missed this but it's important to investigate what happens to the 2OG. Is succinate the only product or is ethylene also produced? This possibility should also be considered in the plant studies, i.e. is there any evidence for responses related to perturbed ethylene metabolism. The authors consider a signalling role relating to AASA/P6C, but seem to ignore a potential ethylene connection.

As stated above, we checked for ethylene production with negative result. EFE produced 6 times more guanidine than the plant enzymes under the same condition, but even 100-fold lower ethylene production would have been clearly detected.

Page 12 - 'plants have been shown to....' Perhaps note how hydroxy guanidine is made?

We now mention the canavanine-γ-lyase that cleaves canavanine into hydroxyguanidine and homoserine.

Overall, I thought the discussion was good, but perhaps a bit long/too speculative on pages 12/13 and this detracted from the biochemical assignment of the enzyme. I'd suggest shortening the discussion somewhat - the precise roles of the enzyme can be the subject of future work. As indicated above, some discussion on potential links to ethylene would be appreciated.

Since reviewer 2 wanted more (speculative) discussion on the role of the 2-ODD-C23 enzymes and there was no detectable ethylene production, we took the liberty to leave the discussion largely unaltered.

I'd also like to see some more consideration/metabolic analyses of guanidine related metabolism in the genetically modified plants.

Such analyses will certainly be included in future experiments once we get an idea about the physiological role of the 2-ODD-C23 enzymes.

Page 16 - mass spectrometry

Corrected.

Please add a structurally informed sequence alignment with EFE and other 2OG oxygenases acting on arginine/derivatives.

An excerpt of the alignment is now presented in supplementary figure S2.

**Reviewer #2 (Recommendations For The Authors):**
I would like to see more discussion in the manuscript about the possible interconnection/roles between 2-ODD-C23 guanidine-producing, lysine- ALD1-Pipecolate producing, and proline metabolism pathways during both biotic and abiotic stresses.

Since we were unable to detect pipecolate in any of our plant samples and also our preliminary results with biotic stress did not produce any evidence for a function of the 2ODD-C23 enzymes in the tested defense responses, we would like to postpone such extended discussion until we find a condition where the physiological function of these enzymes is evident.

Fig. 4: Authors should change colors for Col-0, 0.2 HoArg and ctrl? They look too similar in my pdf file.

We changed the colors in figure 4 and hope that the enhanced contrast is maintained during the production of the final version of our article.